# Interaction Design of the Mixed Reality Application for Deaf Children

Aigerim Kydyrbekova
aigerim.kydyrbekova@nu.edu.kz
L. N. Gumilev Eurasian National University
Astana, Kazakhstan

Nurziya Oralbayeva
nurziya.oralbayeva@nu.edu.kz
Graduate School of Education, Nazarbayev University
Astana, Kazakhstan

Azamat Kenzhekhan
Sultan Omirbayev
azamat.kenzhekhan@nu.edu.kz
sultan.omirbayev@nu.edu.kz
Department of Robotics and Mechatronics, School of
Engineering and Digital Sciences, Nazarbayev University
Astana, Kazakhstan

Alfarabi Imashev
Anara Sandygulova*
alfarabi.imashev@nu.edu.kz
anara.sandygulova@nu.edu.kz
Department of Robotics and Mechatronics, School of
Engineering and Digital Sciences, Nazarbayev University
Astana, Kazakhstan

## ABSTRACT

In recent years, using mixed reality (MR) for teaching sign language has become more feasible. Existing research highlights the emerging importance of MR in teaching by providing a unique way of acquiring information through a visual modality in an immersive environment. This adds a perceived value to the application of MR in tackling the challenges of teaching and learning sign language to deaf and hard-of-hearing (DHH) children. In Kazakhstan, responding to the needs of the DHH children is often overlooked by educational authorities and stakeholders. Central to this research is the exploration of deaf-friendly ways of teaching Kazakh-Russian Sign Language (K-RSL) to DHH children using state-of-the-art MR technology. In this paper, we propose an iterative design framework of a holistic MR system that is tailored to respond to the needs of the DHH children through an engaging vocabulary learning activity co-designed with the participants. To this end, we conducted a pilot experiment with four deaf children aged 6-14 years old, who evaluated the MR application and four different signing avatars. Based on the preliminary observations, we discuss the limitations and possible directions for system improvement.

## CCS CONCEPTS

• **Computer systems organization** → **Embedded systems**; *Redundancy*; Robotics; • **Networks** → Network reliability.

## KEYWORDS

mixed reality, sign language, human-computer interaction, human-robot interaction

*Corresponding author: anara.sandygulova@nu.edu.kz

**ACM Reference Format:**
Aigerim Kydyrbekova, Nurziya Oralbayeva, Azamat Kenzhekhan, Sultan Omirbayev, Alfarabi Imashev, and Anara Sandygulova. 2023. Interaction Design of the Mixed Reality Application for Deaf Children. In . ACM, New York, NY, USA, 6 pages. https://doi.org/XXXXXXX.XXXXXXX

## 1 INTRODUCTION

Mixed reality (MR) is a combination of virtual and physical environments that allows for a more immersive and interactive learning experience. In education, MR technology has gained traction in recent years alongside traditional teaching methods by providing students with dynamic visualizations and simulations, allowing them to explore and experiment in an MR setting [16]. Research demonstrates successful implementations of MR for science education [5], language learning [25] and human-robot interaction [11, 21]. Among the educational benefits of MR, evidence from earlier work suggests that learning in the MR environment can significantly increase students' motivation and cognitive gains [8]. Additionally, MR also provides teachers with new ways to engage students, providing them with opportunities to collaborate, problem-solve, self-direct, and create in a mixed reality [9, 17]. It is especially well-received by inclusive education as it offers great potential for increasing accessibility for students with disabilities, addressing their challenges and allowing them to participate in interactive learning experiences that were previously not possible [6]. Our work sets out to address these challenges by leveraging an MR application in the context of sign language teaching and learning.

Using MR to support teaching and learning sign language has not yet been fully explored, although some scientific evidence exists to date. One of the prominent advantages of MR technology lies in enabling DHH individuals to practice signing with holographic virtual avatars in real-time [1, 18, 23, 27]. Additionally, MR can provide students with visual aids and simulations, making the learning process easier and more engaging [27]. As visual information is often the primary modality for DHH children, MR holds a promising potential to bridge the social and educational gaps in overcoming learning-related limitations of deaf sign language users. Globally, DHH children encounter many barriers to basic education, let alone quality education. Evidence shows that even provided with access

to education, deaf children are still faced with many challenges, such as the lack of natural language input and exposure from early literacy stages, a lack of teachers fluent in sign language, and a lack of a deaf-friendly learning environment aimed at yielding effective learning [19]. These issues place deaf children in a detrimental educational setting having them deal with linguistic deprivation, which subsequently limits their functionality by leading to serious shortcomings in social, academic, and psychological aspects of deaf children's lives [19].

Central to our work is the MR's emerging role in creating a sign language-friendly learning environment for teaching signs to DHH children. The main purpose of this work is thus to teach and learn Kazakh-Russian Sign Language (K-RSL) by providing translations of words in K-RSL and composing sentences using the capabilities of mixed reality. Besides, we also aim to enrich DHH children's vocabulary as well as to improve creative thinking through expressing thoughts and ideas through sentence composition and storytelling components of the proposed learning activity. To this end, we performed a pilot testing with the deployment of cutting-edge MR technology involving four DHH children as co-designers to explore the potential of MR and its acceptability. This paper contributes to the field of human-computer and potentially human-robot interaction domains by establishing a design framework and a rigorous methodological protocol for the application and evaluation of MR systems based on the iterative interaction design.

## 2 BACKGROUND

### 2.1 Mixed Reality in Deaf Education

Despite the attempts toward revealing the potential of MR, its possibilities in teaching sign language have not yet been fully explored. Some prior work leveraged various MR wearable devices and evaluated their capabilities in aiding deaf education. The study conducted by Adamo-Villani and Anasingaraju [1] used a holographic sign language avatar, powered by Meta 1 developer kit and the Unity 3D game engine, for developing K-6 children's math knowledge and skills. Their findings revealed that the system was perceived as more fun and easy to use than anticipated. It was also observed that children enjoyed all activities and reported the signing avatar to be less challenging when following and completing the lesson. Miller et al. [18] evaluated the lecture comprehension of DHH university students using Google Glasses and The EPSON Moverio BT-200. Although the quantitative results were not supportive of the increased comprehension scores as hypothesized, the qualitative responses of the participants point to the feasibility of the glasses in reducing the effect of visual field switches during the lecture. Next, Parton [20] piloted the use of Glass Vision 3D with deaf upper elementary school children for vocabulary acquisition purposes. The results revealed that children perceived Google Glass overall positively compared to other devices (e.g., iPad and iPhone). More recently, Shao et al. [23] evaluated the efficacy of mixed-reality-based interactive motion teaching with 60 participants by comparing the proposed system with a desktop-based non-interactive baseline. They utilized the HTC VIVEPro and a ZED Mini 3D stereo camera. The results in terms of the system's usefulness revealed some statistically significant improvements in children's learning of American Sign Language (ASL) signs.

So far, there has only been a limited attempt in utilizing a more sophisticated holographic wearable device. Recently, Yang et al. [27] developed a HoloLens-powered MR application and conducted a mixed-subject design experiment to evaluate its usability with eight hearing participants fluent in ASL. Since the study relied on hearing participants instead of DHH individuals, the results cannot be generalized to the DHH community. Yet, it can be implied from the results that close consideration should be given to sign language users' preferences about the settings and appearances of the MR application and its components, such as a human-like avatar.

### 2.2 Signing Avatars

Signing avatars – human-like virtual characters – are becoming increasingly popular as an assistive technology facilitating communication for DHH students [28]. Generally, research findings support the value of signing avatars acknowledging their benefits in aiding to overcome challenges often encountered by the DHH students and making information accessible [3, 7, 15]. Among earlier educational applications of signing avatars are Science and Math in an Immersive Environment (SMILE) and Mathsigner developed for teaching math and science concepts to DHH children [2], SignTutor [4], and CopyCat [26] that allow students to practice new signs while providing feedback to improve the students' signing abilities. Evidence suggests that students generally improve their receptive, expressive, and working memory abilities when learning from the signing avatars [26]. However, there exists evidence counter to what was established about signing avatars' effectiveness and understandability. Some studies found the recordings of a human sign interpreter to be preferable to the avatar [14, 22]. This implies that more research should be dedicated to the robust production of the manual as well as non-manual signals necessary for the delivery of visual information.

Moreover, one of the major limitations currently faced by the researchers of signing avatar technology is the lack of a standardized protocol for the subjective evaluation of signing avatar systems. Frequently, scholars refer to questionnaires in written language [12], videos [14], and picture books [10]. Although useful, the deaf-friendliness of these evaluation practices is still debatable. A possible solution to this issue can be observed in [1] and [13], who introduce a visual toolkit for measuring user acceptability and satisfaction with the sign language animation. Similarly, we aim to establish a subtle evaluation protocol for user studies involving DHH children. Considering the context of Kazakhstan, the DHH children often lack the literacy skills required for their knowledge construction as they receive little to no exposure to their native sign language or to written language. Therefore, using a variety of tools, such as object and image sorting tasks, would be a hands-on solution to present questions for the evaluation of the virtual avatar's animation from users' perspectives.

### 2.3 Participatory Design

To establish a deaf-friendly and age-appropriate measurement protocol towards the evaluation of virtual systems, it is paramount to involve the DHH children in the process of system development. This is normally accomplished through a user-centred design

approach, often referred to as participatory design (PD) for developing, prototyping, and refining a system in collaboration with the target population [24]. Participatory design, otherwise known in the SL research domain as community-engaged research (CEnR), has seen recognition in different areas of sign language research [24]. However, their application seldom ensures the perspectives of the Deaf are articulated decently through the use of proper data collection tools. Therefore, establishing a common protocol to be easily followed by the deaf or hard-of-hearing individuals as both researchers and participants should be prioritized.

## 3 METHOD

As part of the interaction design (ID) process, we take advantage of Participatory Design (PD). As is characteristic of any ID, the process includes a few cycles of activities such as identifying needs, establishing requirements, developing a number of prototypes that meet those requirements, and subsequently evaluating these prototypes by users and other stakeholders.

In line with Spinuzzi [24], we followed the basic stages of PD research which encompasses initial exploration, discovery process, and prototyping combined with the constituents of the interaction design processes mentioned above. The entire process thus consisted of the following:

- **Cycle 0:** Initial exploration. This cycle encompassed the investigation of the background research, a close literature review for methodological frameworks, establishing requirements, understanding the participants, designing a learning task, and obtaining ethical approval. The primary goal of the MR system was to stimulate cognitive gains while engaging children in game-based activities. Therefore, the first step was to understand the participants' needs in dealing with new signs and concepts by taking into account the cognitive processes involved and the cognitive limitations of the users by age and individual differences. As such, the system needs to: (1) establish a sign language-friendly learning environment by bringing the immersive reality and signing avatars to the center of DHH children's learning experience; (2) be accurate and reliable in interpreting sign language gestures; (3) provide real-time feedback to the children as they practice signing; (4) utilize the wearable device's built-in functionality of head movement in response to face tracking.
- **Cycle 1:** Prototyping the system for the MR application with the participants of the targeted age group. This cycle went through several iterations and improvements from paper to software prototypes based on participants feedback to ensure accessibility, provide engaging and interactive environment for learning. While creating a prototype, the list of potential learning tasks was used to identify the particular skills and concepts that the application aims to impart.
- **Cycle 2:** Implementation of the application. The primary objective of this cycle was to select appropriate hardware and software, and make use of all the available software features to develop the MR application meeting the requirements identified in previous cycles. During this cycle, the final learning objectives were determined, the suitable technology was selected, and the application was created and tested.

- **Cycle 3:** Evaluation of the MR application through using hands-on physical objects to facilitate the presentation of difficult concepts (e.g., natural, humanlike, etc.) to DHH children. At a certain stage of the process, an assessment was conducted to determine if the application is achieving its intended learning goals and offering benefits to the age group it is intended for. This evaluation was done over a period of time to measure the application's long-term effectiveness.

These design principles were discussed with the professional sign language interpreter and educator, who approved using them in the implementation of our MR-based system.

### 3.1 Learning Task

We developed a learning task in the MR environment for the acquisition and memorization of signs corresponding to different words. Children are immersed in the mixed reality environment where they see 3D objects and videos representing target words (e.g., an animal, an electronic device, Instagram, etc.), two virtual humanlike avatars, a human interpreter, and a stick figure all signing the target word.

The scope of this learning task encompasses the following areas:

- gaining knowledge and comprehension of the meanings of words in either Russian or Kazakh;
- ensuring accurate spelling of words in Russian or Kazakh;
- remembering translations of specific words into Kazakh-Russian sign language demonstrated by multiple signers;
- developing the ability to recall and memorize the signs by constructing sentences using the words.

Having a 3D representation of a subject with interactive features, such as zooming, rotation, and the ability to manipulate elements, provides participants with a more complete understanding of the topic with visualization and spatial awareness. By having the ability to inspect the object from different angles and interact with its components, participants can gain a better grasp of the subject. By allowing participants to see and interact with a visual representation of the word or concept, they can gain a deeper understanding and retain the information more effectively. At this point, the learning task for comprehending the definition of a word and having a full understanding of it could be fulfilled.

The task of accurate spelling was a secondary goal. Providing the names of things and events near objects reinforces the memory to recall or learn their correct spelling in Russian or Kazakh.

The primary objective of the learning task is to help DHH children acquire and retain signs, and the translation of words into the Kazakh-Russian sign language is performed by various sources including a video of a real signer, two different computer-generated avatars, and a representation using a stick figure to demonstrate the sign. Also, the presence of several options for reproducing signs makes it possible to evaluate the advantages and disadvantages of each and use the best option in the next iterations of the project.

The final step in engaging with the system involves a test where the child can assess their own recall of the words and gestures they have learned, and it also provides them with a chance to contemplate what they have covered. During this quiz stage, the child is free to construct sentences using the words they have previously learned. The subject matter or theme for the sentences

is left to the child's own preference, allowing them to use their imagination and creativity to craft their own story. Children were awarded points if they were able to construct a sentence that follows the guidelines of the Kazakh-Russian sign language system.

## 3.2 The Proposed System

We developed a mixed-reality system integrating the HoloLens 2 headset that enables the acquisition of vocabulary knowledge through an immersive environment. The development of the system followed an iterative process, meaning it went through several cycles of design, implementation, and evaluation towards the establishment of a design framework for a robust and scalable application.

**Microsoft HoloLens 2.** For the best MR experience, we have chosen HoloLens 2. In the very first version of the program HoloLens 1 has been used, with additional eye tracking components, however, after discovering that HoloLens 2 has eye tracking by default, it was decided that it is more convenient to use it. Moreover, HoloLens 2 is known for exceptional hand tracking, as well as better Software applications such as MRTK.

**The first prototype:** The initial software prototype focused on teaching Kazakh-Russian sign language through a demonstration and repetition of nine phrases. These phrases included "Do you like football?", "I am a housewife/househusband", "I have a cat", "The weather today is humid", "Do you watch YouTube?", "What is your favorite movie", "Do you have Instagram?", "What's your phone number?", and "Do you like rain?" The demonstration was done through two computer-generated avatars, one being a sign-language interpreter and the other a stick-figure signing the same phrases. Notably, the prototype did not include any options for interacting with 3D objects. The first version was used to refine and improve the subsequent iteration of the model. Prior to testing the model with deaf and hard-of-hearing children, it was first tested with a hearing child to identify key takeaways and improve the model. During the interaction, several important points were highlighted, including:

- The height difference between researchers and children was problematic. The calibration process, which was done based on adult measurements, resulted in buttons that were difficult for children to access. To address this issue, adjustments were made to the height of the camera and buttons in the 3-D space to ensure they were appropriate for the participants;
- Initially, the plan was to compare different sign language avatars side by side to determine the best one. However, it became apparent that having multiple avatars created confusion, distracting the participants. Therefore, it was decided to present only one avatar and one sentence at a time to eliminate distractions;
- To accommodate the relatively long duration of the experiment, the model was modified to allow participants to remain seated throughout the session;
- The main menu was redesigned to enable children to return to the beginning and start over as needed.

The initial prototype was tested with DHH children, and after gauging their sign language proficiency, deaf educators and interpreters recommended that the project begin with teaching

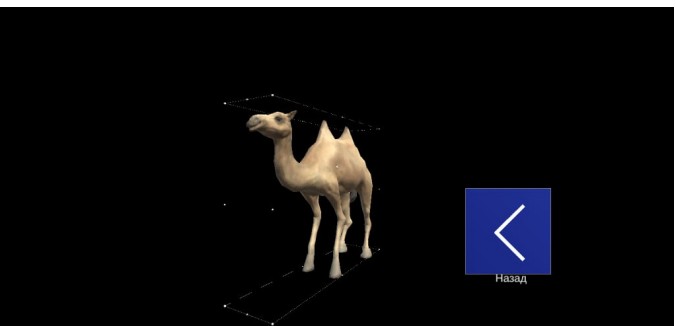

**Figure 1: A screenshot of the MR environment**

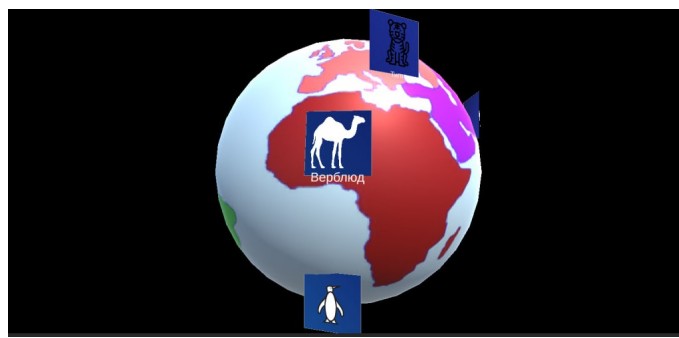

**Figure 2: A screenshot of the MR environment after the second iteration**

individual words rather than phrases. As a result, the decision was made to focus on separate words, utilizing 3D representation to take full advantage of the capabilities of Mixed Reality in visualization and spatial awareness. Once the children have learned the signs for the target words, the next step will be to move onto phrases and sentence composition using those words.

**The second prototype:** The latest version of the MR (mixed reality) application features a globe that has various animals attached to different countries or continents, which act as buttons. When a user clicks on one of these buttons, they are taken to a scene that showcases a 3-D animal model, informative text about the animal, sign language writing, and a video displaying the sign. Target words are names of animals in Kazakh and Russian languages. Additionally, users can interact with the animal model, and there are different food/snacks available for the animals to eat.

Previously, the application had been connected to the HoloLens Device Portal to enable users to see what they were viewing, and audio cues were added for situations where a PC was unavailable.

Once the user has learned the target words with the animal model, the next scene is a quiz where they can test their knowledge by constructing sentences and phrases using the target words. To enhance user experience, a swipe mechanism has been added to allow users to change the scene effortlessly, since buttons may not always be reliable.

## 3.3 Evaluation

We conducted a series of evaluations following the iterative cycle of designing the system. Thus, there have been three evaluations: two evaluations in the laboratory settings, and one evaluation on site of an inclusive school.

*3.3.1 Recruitment.* This pilot research followed rigorous ethical reviews prior to being approved. It received approval from the Institutional Research Ethics Committee (IREC) at Nazarbayev University. Video translations of written child assent and informed consent forms in the local sign language were distributed to all children and their parents by means of messaging. The researchers ensured the provision of a brief and simplified explanation about the aim and the procedures of the study.

*3.3.2 Participants.* We recruited four participants (F=2, M=2) between the ages of 6 to 14 years old from a local mainstream secondary school, fostering inclusive education. All participants were diagnosed with bilateral sensorineural hearing loss, and communication without the use of sign language or hearing aid is almost impossible for them. Only one participant had an average familiarity with sign language, while the other participants had limited sign language proficiency. The latter participants were not able to self-assess their knowledge of sign language. Deaf educators and interpreters rated their level of sign language as low and domestic using their own home sign language.

*3.3.3 Requirements for the setup.* The researchers were allocated a separate room that served as an extension of a regular classroom where deaf children would normally sit alongside their hearing peers. One of the researchers set up the room to create an MR environment with a HoloLens 2 headset and a laptop that controls the MR application. As the main purpose of the MR system was to teach signing to deaf children and evaluate its user acceptability, the MR application integrated four different agents performing signs, such as a professional human sign language interpreter, a stick figure, and two signing avatars, one of which is custom-designed and the other is a publicly available avatar. The interaction with the system consists of a series of events that occur sequentially, which are categorized as before, during, and after the main interaction. Before the interaction, the participant enters the classroom, receives a brief introduction to the application, answers demographic questions, completes a pretest questionnaire, completes pre-interaction mood test and learns how to use the HoloLens device. During the interaction, the participant puts on the HoloLens glasses, performs a calibration process, watches video instructions in sign language, launches the application and selects an animal on a globe. The system then displays a 3D image of the animal and a video that includes sign language translation and signwriting. The participant can interact with the objects, read information about the animal, repeat the sign, and return to the main menu to select another animal. After interacting with all the animals, the participant can take a quiz where he or she constructs sentences using sign language target words (signs). The system checks the accuracy of the sentences and gives points for correct answers. The interaction ends when the participant constructs sentences using all available words or decides to end the session. After completing the interaction, the participant writes a post-test questionnaire, and participant completes the post-interaction mood test.

## 4 FINDINGS

As was mentioned earlier, the process of designing the MR application was iterative and went through three cycles of programming, implementing, and piloting the system with children of targeted age groups and redesigning the learning activities. Drawing on this experience, we highlight the following limitations and key takeaways observed from each iteration:

*First in-lab evaluation.* Our first implementation and evaluation of the MR application with a hearing child were conducted in the laboratory setting. The results of our observations revealed some serious issues related to the design of the MR environment and its ergonomics. In particular, the participating child reported having trouble with pressing the button in the mixed reality, which caused some major inconveniences in making the interaction smooth. This is most likely to affect the user's overall perception of the system's usability. Therefore, in the next iteration, the button should enable the command by touch, pitch, and pressing more easily. Besides, the in-the-lab setting has proven to be lacking some ergonomics, meaning the arrangement of the experimental setup room should meet the system requirements and allow enough space for moving around during the interaction. Initially, we planned that the child would be in a standing position, however, it was found during the first piloting that the seating position would be more convenient due to the relatively long duration of the experiment.

*In-the-wild evaluation.* Based on our initial observations from the piloting of the MR application with the four DHH children, we identified some discrepancies not only in the proposed MR application but also in the design of our experiment. One of the major challenges faced by both the researchers and sign language interpreters was the children's little to no proficiency in sign language. Even when children had proficiency to some extent, their levels varied vastly. Some of the children had custom sign language developed within one's home conditions and comprehensible only to the family members. As a result, the sign language interpreters often struggled to establish a connection with those children. Since most of them had a certain degree of literacy in written language, providing written text accompanying the signs would come in handy at the next in-the-wild implementation stage. We collaborated with the educators and vice principal of the school to discuss our design decisions and received useful feedback on how to approach the DHH children with varying levels of sign language command, who granted their consent to participate in the study. In addition, the vice principal, who held expertise in teaching the Deaf, guided us on how to design the learning task.

As regards the learning task, initially, children did not have the freedom of choosing which signs to learn and the order in which the signs appeared. Later, during the second interaction, we enabled the option to choose and return to the main menu with signs so that a child can enjoy the opportunity to practice the new signs by choosing their order. Moreover, we introduced a sentence composition task, which also caused some unanticipated hurdles. It turned out during the evaluation that the DHH children struggled to build sentences from signs due in part to not being familiar with

                                              

the sentence structure in the local sign language. Another potential explanation for this may be the introduction of independent signs out of context, that is, all signs should be introduced promptly so that a child can make sentences.

Other considerations for further improvement of the MR application include the time on task (interaction time). We assume that the more time the child spends with the MR application, the more effective it can be for increasing the learners' sign language vocabulary knowledge. It is important to note that one of the impeding factors, in this case, was that we had to rely largely on the sign language interpreter to guide the child during the interaction steps. This provides us with an implication for the independent usage of the MR application by the child. A potential solution would be making and presenting an introductory video explanation for the child to understand how to use and operate the MR application. Besides, we observed the difficulty following the child's actions in the mixed reality. To facilitate this, we propose to add a sound effect to allow the researchers to identify whether the child has pressed the button to proceed or finish the interaction.

*Second in-lab evaluation.* After the second in-the-lab evaluation, we found some other mismatches in the system. For example, the distance between the child and the mixed reality objects turned out to be unreachable. Therefore, it is important to reduce the field of view to bring the objects closer to the child. Another consideration for future implementation is to integrate eye-tracking to collect behavioral data from the interaction.

## 5 CONCLUDING REMARKS

As the next step of our system and framework design, we plan to implement the improved MR application in the wild with the DHH children. Therefore, we plan to further fine-tune the system by integrating the above-mentioned considerations. In future work, we also aim to focus on sign language creating a sign language learning scenario within HRI by employing a social humanoid robot and/or a teleoperated robot alongside the MR application to compare their effectiveness for deaf children's cognitive and emotional gains.

## 6 ACKNOWLEDGMENTS

We would like to extend our gratitude to the professional sign language pedagogues and interpreters, whose guidance was crucial, and the students who participated in the study. This work was supported by the Nazarbayev University Collaborative Research Program grants: 091019CRP2107 and OPCRP2022002.

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
