# OpenReview forum: "Interaction Design of the Mixed Reality Application for Deaf Children"
_humanrobotinteraction.org/HRI/2023/Workshop/VAM-HRI — VAM-HRI 2023 Oral_

### Official Review · Program_Chairs · 2023-02-24
**Accept**

**Rating:** 7
**Confidence:** 5

**Review:**

Reviewer 1:

The authors talk about the user studies for mixed reality applications for deaf children to teach them to sign language. The article presents an interesting approach to using MR for educational purposes as well as interesting user studies.

Potential improvements:

There is no robot in the loop and while the application is very interesting, it is hard to see the direct application of these methods to human-robot interaction. I think you can extend that whit e.g. robots teaching kids sign language together with Mr.

It is a bit hard to see/understand how the framework works just from the images. It would be better to attach the video recorded while interacting with the interface.

A very low number of participants (4). I wanted to argue that there is no statistical analysis but it is hard to analyze anything with 4 participants. More participants and statistical analysis should be added in the final version of the article

Reviewer 2:
This paper investigates the usage of MR devices for teaching deaf and hard-of-hearing children K-RSL. While this paper is very interesting and addresses an important problem, the discussion on integration with a robot is a bit lacking. Overall I recommend this paper be accepted, but I encourage the authors to include some of their thoughts on how they plan to integrate their MR system with a social robot/teleoperated robot in more detail in the paper, and also when they present at the workshop.

---

### Decision · Program_Chairs · 2023-03-02

Accept (Oral)